# Kidney Bean Substitution Ameliorates the Nutritional Quality of Extruded Purple Sweet Potatoes: Evaluation of Chemical Composition, Glycemic Index, and Antioxidant Capacity

**DOI:** 10.3390/foods12071525

**Published:** 2023-04-04

**Authors:** Eny Palupi, Nira Delina, Naufal M. Nurdin, Hana F. Navratilova, Rimbawan Rimbawan, Ahmad Sulaeman

**Affiliations:** Department of Community Nutrition, Faculty of Human Ecology, IPB University, Bogor 16680, Indonesia

**Keywords:** extrusion, fiber, glycemic index, indigenous crop, legumes

## Abstract

The extrusion process may influence the nutritional profiles of carbohydrate-rich food ingredients, including the glycemic index (GI) and antioxidant capacity. This study aimed to evaluate the nutritional profile of extruded purple sweet potato (EPSP) substituted with kidney bean flour (KBF) (0, 30, and 40%). These foods were further characterized by their proximate composition, resistant starch, polyphenols, GI, and antioxidant capacities. The 40% KBF substitution enhanced the protein and fiber contents of the EPSP by up to 8% and 6%, respectively. Moreover, it also revealed that EPSP with 40% KBF substitution had a low-GI category (53.1), while the 0 and 30% substitution levels had a high-GI category, i.e., 77.4 and 74.7, respectively. However, the extrusion processing reduced the anthocyanin content and antioxidant capacity of purple sweet potato flour containing 40% KBF by 48% and 19%, respectively. There was a significant relationship between the GI values of proteins, fats, and fibers (*p* < 0.05). The insignificant effect of resistant starch and phenol contents on GI value was recorded due to the low concentrations of those components. KBF substitution could ameliorate the profile of protein, fiber, and GI, but not for antioxidant capacity. The other innovative processes for preserving antioxidant capacity might improve the product quality.

## 1. Introduction

Diabetes and obesity have been the primary health problems worldwide. The International Diabetes Federation recorded that 537 million adults, or more than 10% of the world population, suffer from diabetic mellitus type 2 [1]. This number is predicted to increase to 783 million in 2045. The epidemic’s prevalence demands a higher level of responsibility from many stakeholders worldwide to find solutions, or at least slow down its rate. A high glycemic index (GI) diet with poor fiber and other essential nutrients may lead to a high risk of diabetes [2]. In addition, stress-oxidative and low physical activities could speed up the path to diabetes [3]. Many efforts have been made to slow down those processes. One of them is by developing low-GI foods that are easy to serve, have the desired nutritional value, and are preferred by the community.

Purple sweet potato (*Ipomoea batatas* L.) is a tropical root that is easily cultivated and contains a significant amount of fiber (2.3–3.9%) with a low-medium GI (54–68) and strong antioxidant activity (IC_50_ 65.35 ppm) that might be beneficial in preventing diabetes and other non-communicable diseases [3,4,5,6]. Among other sweet potatoes, the deep purple sweet potato has the highest anthocyanin and antioxidant capacity [5]. However, the short shelf life and complicated processing method hinder the broad utilization of this tuber. To improve its shelf life and acceptability, purple sweet potato might be processed into flour and extruded products by using a high-temperature, short-time technique. However, this high-temperature process might transform this commodity from a low-GI food into a high-GI food and lower its antioxidant capacity [7].

Extruded foods have been a topic of discussion in the food industry for many years and continue to be a popular and convenient food choice for many consumers. Some concerns have been raised regarding the nutritional quality and safety of extruded foods [8,9]. Extruded foods are made by processing a mixture of ingredients, such as grains and legumes, under high pressure and temperature to create a dough-like material that is then pushed through a die to give it a specific shape. One concern with extruded foods is that the high heat and pressure used during processing can cause a loss of nutrients, especially vitamins and bioactive compounds [8,9]. Additionally, some studies have suggested that certain compounds formed during the extrusion process may have negative health effects. Some alternative approaches for handling this issue are low-temperature extrusion and legume substitution in the mixture [10,11].

Substitution of legumes has been acknowledged to offer various benefits, i.e., protein enrichment, texture improvement, satiety improvement, and positive alteration of the glycemic profile, lipidemic profile, and gut microbiota [12,13]. Legumes contain significant amounts of protein, fiber, and various secondary compounds, like phenolic compounds, that might provide various health benefits [14]. Indonesia preserves abundant biodiversity, including legume diversity [15]. The present research utilized kidney bean (*Phaseolus vulgaris* L.) as a main ingredient substitution in developing extruded purple sweet potato (EPSP). This legume was selected since it has the lowest GI value (26), with a significant amount of fiber (5.29%) and protein (24.79%) [16]. Furthermore, it is adaptable to being cultivated at high temperatures with low water intake [17].

This study aimed to evaluate the effect of kidney bean flour (KBF) substitution in the EPSP on nutritional quality, including macronutrients, GI function, and antioxidant capacity. Other important parameters that might influence the GI value were also observed in this research, i.e., polyphenol content and resistant starch [18]. This study is expected to formulate an extruded, ready-to-eat food containing significant fiber and having a low GI made from purple sweet potatoes and kidney beans. The final purpose of this research is to develop a nutritious product that is well-suited for promoting and maintaining the nutritional status and health of the community.

## 2. Materials and Methods

This research has obtained permission from the Commission on Research Ethics Involving Human Subjects, IPB University Number: 661/IT3.KEPMSM-IPB/SK/2022.

### 2.1. EPSP Processing and Nutrient Analysis

The EPSP was made from purple sweet potato flour (PSPF) and kidney bean flour (KBF) with three formulas, i.e., 100% PSPF (F0, Formula 0), 70% PSPF and 30% KBF (F1), and 60% PSPF with 40% KBF (F2). These PSPF and KBF combinations made up 76% *w*/*w* of the total formula. The PSPF and KBF were produced through the following processing stages: cleaning, steaming (100 °C, 20 min), double drum drying (70–120 °C), and cabinet drying (60 °C, 2 h). The additional ingredients (% *w*/*w*) included rice flour (12.46), cornstarch (1.6), powdered milk (0.9), palm oil (1.8), water (7.2), and emulsifier (0.046) (Table 1). All these ingredients were mixed and then processed using a twin screw extruder at a temperature of 60 °C, an auger speed of 40 Hz, and a screw speed of 40 Hz. The resulting extrusion product is shown in Figure 1. The proximate analysis was performed to measure the contents of water, ash, protein, fat, and carbohydrates with the methods of oven-gravimetric, dry-ashing, Kjeldahl, soxhlet, and by-difference, respectively. Dietary fiber content was also measured using the enzymatic gravimetry method.

### 2.2. Polyphenol Content Analysis

The polyphenol analysis was performed based on Jayanegara et al. [19]. A total of 0.2 g of sample was added to 10 mL of methanol (50%) and put in an ultrasonic water bath for 20 min at room temperature. Then it was centrifuged for 10 min at 3000× *g* at 4 °C. Then, a 0.2 mL aliquot of sample extract was added with 1.25 mL Folin-Ciocalteu reagent and 6.25 mL sodium carbonate, and distilled water was added so that the volume reached 10 mL, was vortexed, and was recorded at 725 nm. Non-tannin phenol was analyzed using 0.1 g of Polyvinylpolypyrrolidone (PVPP) and inserted into a centrifuge tube to separate the content of tannin and non-tannin. 1.0 mL of distilled water and 1.0 mL of sample extract were added, then vortexed and stored at 4 °C for 15 min. After that, it was vortexed again and centrifuged (3000× *g* for 10 min). The supernatant containing only non-tannin phenol was collected. Then 0.4 mL aliquots were taken and added along with 1.25 mL Folin-Ciocalteu reagent and 6.25 mL sodium carbonate. Then, distilled water was added so that the volume reached 10 mL. Total phenol and total tannin were calibrated against gallic acid solution as standards, and values were expressed as mg gallic acid equivalents (GAE).

### 2.3. Resistant Starch Analysis

Resistant starch analysis was performed based on the AOAC 2002.02 method and the AACC 32–40 method available from the Megazyme International Protocol [20]. A total of 0.1 g of sample was mixed with 3.5 mL of sodium maleate buffer (pH 6.0) in a 37 °C water bath for 5 min. Then, 0.5 mL of pancreatic amylase enzyme was added to each sample tube. They were placed in a shaking water bath horizontally at 37 °C for 4 h (200 strokes/min), added with 4 mL of 95% ethanol, and then vortexed and centrifuged (4000 rpm, 10 min). The pellet was added to 2 mL of ethanol (50% *v*/*v*) and vortexed. Then, 6 mL of ethanol was added (50% *v*/*v*), shaken, and centrifuged again (1500 rpm, 10 min). Pellets were added to 2 mL of 1.7 M NaOH and shaken for 20 min. After that, 8 mL of sodium acetate buffer (1 M, pH 3.8) was added to each tube and shaken again. Then, 0.1 mL of AMG solution was added immediately and incubated for 30 min. Take 1.5 mL of the aliquot, then centrifuge (4000 rpm, 5 min); 0.1 mL of the supernatant was transferred to a glass test tube, added with 3 mL of GOPOD reagent, and incubated (50 °C, 20 min). The absorbance was read at 510 nm.

### 2.4. Glycemic Index Testing Procedure

The GI testing procedure is referred to as the ISO 26642 [21] method. The ISO 26642 mentions that the GI should be determined based on the average glucose response of a minimum of 10 healthy subjects [21,22]. The subjects in this study were recruited using a purposive sampling method. The inclusion criteria for the subjects were that they were male or female in the age range of 18–30 years, had a normal Body Mass Index (BMI) of 18.5–24.9 kg/m^2^, had no allergies or food intolerances, had a normal fasting blood sugar value (<110 mg/dL), had a healthy condition as stated by a doctor, engaged in light to moderate physical activity, and were willing to have their blood glucose levels measured. Exclusion criteria for the subjects included having a history of diabetes mellitus, being pregnant or breastfeeding, experiencing digestive disorders, consuming alcohol, undergoing medication, and smoking [21].

The GI value in this study was assessed in vivo with 13 participating subjects, including seven males (53.85%) and six females (46.15%) (Table 2). The nutritional status category [23] showed that all subjects were in the normal nutritional status category (BMI = 21.13 ± 1.40 kg/m^2^). Statistical tests showed that the data on height, weight, and BMI were not significantly different (*p* > 0.05). So, it can be said that the subjects involved in this study were homogenous.

The reference food was 25 g of D(+)-Glucose anhydrous for biochemistry (Merck) that was dissolved in 250 mL of water, which was repeated twice [21,22]. The test foods were three formulas of extruded purple sweet potato and kidney bean, i.e., F0 (100% PSP flour), F1 (70% PSP flour and 30% kidney bean flour), and F2 (60% PSP flour and 40% kidney bean flour). Based on ISO 26642 [21] about the determination of the glycaemic index (GI), in Section 2.2 titled “Carbohydrate Portion”, it is stated that the GI values could be determined by the weighed portion of food containing either 50 g of glycaemic carbohydrate or, if the portion size is unreasonably large, 25 g of glycaemic carbohydrate may be used. The evaluated samples have a very low specific gravity, so they have a high density (volume per weight). This high volume per weight made the 50 g glycemic carbohydrate equal to 82 g of product (about two bowls), which is unreasonable to be consumed within 12 min. Therefore, the study used 25 g of available carbohydrate instead of 50 g of available carbohydrate. The extruded products that must be consumed by the subjects were 33, 37.7, and 41 g for F0, F1, and F2, respectively. The test food was served in a plastic bowl with 250 mL of water.

The blood glucose was measured by taking a sample of the subject’s blood using the finger-prick capillary blood sample method with the Accu-Check Performance Glucometer. The subjects underwent complete fasting (except for water) for approximately 10 h (22:00 p.m. to 8.00 a.m. the next day). Before they were given intervention food, their blood samples were taken at minute 0 to determine fasting blood glucose levels. The subjects were required to finish the food provided for 12 min [21,24]. Blood samples were taken at 15, 30, 45, 60, 90, and 120 min. The time interval for giving the test food is five days.

Data on blood glucose levels at each time was plotted into a graph of time (x) and blood glucose (y). The GI value was calculated by comparing the area under the curve between the test and reference foods. The method used to calculate the Incremental Area Under the Curve (IAUC) was the trapezoid rule [21,22]. The method measured the area above the baseline by ignoring the area under the curve. The glycemic index values of the subjects were then averaged to obtain the food’s GI value.

### 2.5. GI for Mixed Food and Glycemic Load Assessment

Suggestions for serving the extruded were adjusted to match the suggestion for serving commercial cereal products in general, which is as much as 35 g and served with 125 g milk. The value of a meal’s Glycemic Index was calculated mathematically using the weighted average GI values for each individual ingredient based on its contribution to the overall available carbohydrate content [24]. The glycemic index (GI) is a system that ranks foods based on how they affect blood sugar levels. The GI values are classified into three categories: high GI (≥70), medium GI (55–70), and low GI (≤55). Another metric used to measure the impact of a food on blood sugar levels is the glycemic load (GL), which takes into account both the GI of the food and the amount of carbohydrates in a typical serving. The GL categories are defined as follows: high GL (>20), medium GL (11–19), and low GL (<10) [21]. The formula for GL is [GI of food × carbohydrate content (g) in one serving size] × 100.

### 2.6. Antioxidant Capacity Analysis

The capacity of the antioxidant products was analyzed by the 2,2 diphenyl-1-picrylhydrazyl (DPPH) method, which is a free radical compound. The principle of this method is to measure the concentration of the sample needed to counteract the DPPH free radicals. The DPPH method consisted of several stages: the extraction of the sample, preparation of standard vitamin C solutions, preparation of DPPH solutions, and analysis of the antioxidant. The antioxidant capacity was expressed as the IC_50_ value and the AEAC (Ascorbic Acid Equivalent Antioxidant Capacity). An analysis of antioxidant capacity was carried out on PSP flour and selected formulas.

### 2.7. Statistical Analysis

The data were analyzed using the Analysis of Variance test (ANOVA) followed by Duncan’s test at alpha 0.05. The data on antioxidant capacity was statistically tested using a *t*-test at alpha 0.05. The relationship analysis among the variables was performed using the Pearson correlation test. All those statistical analyses have been performed using IBM SPSS statistical software version 23.

## 3. Results and Discussion

### 3.1. Chemical Composition

The chemical composition of the developed formulas is shown in Table 3. The legume substitution significantly increased the protein, fat, dietary fiber, water-insoluble fiber, and non-tannin phenol. The F2 with a 40% KBF substitution had the highest content of protein (13.33%), fat (0.87%), dietary fiber (16.31%), water-insoluble fiber (13.75%), and non-tannin phenol (4.74 mg GAE/g) compared to the other formulas (F1 and F0). There were no significant differences in the levels of resistant starch and total phenol among the formulas. Meanwhile, the levels of water-soluble fiber and tannin phenol were reduced with increased kidney bean substitution.

An increase in protein and fiber contents is preferable in terms of the nutritional quality of products. A low-fiber diet has significantly contributed to the growing number of non-communicable diseases [25,26]. Moreover, muscle mass and fat mass in the body complement each other. Lack of protein consumption can trigger a faster decrease in muscle mass, which ultimately increases fat mass and might trigger the metabolic syndrome [27,28]. In addition, muscle mass affected by protein intake also produces cytokines and myokines, which play a role in sugar metabolism [29]. The same preferable profile with the higher non-tannin phenol was also observed, which might improve the health-related roles in the body with less impact as an anti-nutritional compound since the non-tannin type has low reactivity to interfere with other nutrients [30]. Polyphenols in the diet have various health benefits. These compounds are associated with improved lipid profiles, blood pressure, insulin performance, and even the gut microbiome [14,31]. Polyphenols are also reported to lower cardiovascular health risks (i.e., myocardial infraction, stroke, and diabetes) [31].

The findings indicated a significant increase in the content of non-tannin phenols in extruded PSP with increased KBF substitution. This phenomenon is hypothesized to be due to the conversion of tannin phenols into non-tannin phenols under high temperatures and intense friction conditions during the extrusion process [32]. The process appeared to alter the characteristics of the key functional groups of the phenolic compound into non-tannin phenols. This process seems to reduce their ability to bind to proteins, as evidenced by the PPVP test. The conversion of tannin phenols into non-tannin phenols may be attributed to the cleavage of the hydrolysable tannins, which are more susceptible to thermal degradation [32]. Therefore, the extrusion process can lead to changes in the composition and functional properties of PSP, as well as alterations in its bioactive components. The increase in KBF substitution seems to have enhanced the above reaction. However, the amount of tannins in the extruded tuber in this study has not been able to have an effect on the GI value [18]. A complete discussion regarding this correlation is thoroughly reviewed in Section 3.3.

### 3.2. Glycemic Response and Index

The glycemic responses of subjects to reference, F0, F1, and F2, can be seen in Figure 2. The glycemic response of the subjects to the extruded food was below the pure glucose response curve (Food Reference 1 and 2). The difference in kidney bean flour substitution levels is thought to affect the decrease in blood glucose response. The content of protein, fat, and fiber can affect blood glucose responses [18,33]. Peak blood glucose levels in the F0, F1, and F2 groups occurred at 45 min, with an average blood glucose level of 134 ± 5.3, 140 ± 7.4, and 121 ± 3.6 mg/dL, respectively. After that, blood glucose levels decreased gradually and returned to levels that were not much different from fasting blood sugar. The F2 curve has the lowest curve compared to the F0 and F1.

The GI values of the three formulations are presented in Table 4. The result indicates that the drying and extrusion process produced a high-GI product (77.41) on F0 (0% kidney bean substitution). This sample has a higher index than the GI of steamed PSP (66.67), which did not undergo the drying and extrusion processes (Figure 3). The drying (in making PSP flour) and extrusion involve the transformation of starch and protein into a restructured and textured food. The cutting process, coupled with high temperature and low moisture content, causes large molecules of starch to be dextrinated and broken down into shorter chains to have greater solubility in water [34,35]. These processes reduce the particle size, making the starch surface area larger and easier to digest and absorb. In addition, the extrusion process causes the gelatinization of starch [36]. In addition to being caused by heat, starch gelatinization is also caused by pressure and friction. The level of gelatinization increases with higher water content, processing time, and temperature [35].

Moreover, this study also elucidated that 40% of legume substitution in the developed formula (F2) might lower the GI into a low category (53.11a). In contrast, the F0 (77.41b) and F1 (74.68b) belonged to the high category (Table 4). The higher protein, fat, and fiber in the kidney bean might be the primary causative factor for this elevation [37]. Chemical composition and nutrients like the content of protein, fat, dietary fiber, soluble fiber, fiber insoluble, resistant starch, and polyphenol profile (total phenol, tannin, and non-tannin) are internal factors that can affect a food’s GI [18,33,38,39,40,41].

### 3.3. Correlation between Chemical Composition and Glycemic Index

The correlation test results showed a significant relationship (*p* < 0.05) between the GI and the content of protein, fat, dietary fiber, soluble fiber, and insoluble fiber in formulas (Table 5). A high protein content will result in greater gastric inhibitory peptide (GIP) and insulin response, resulting in lower postprandial glucose peaks and reduced glycemic response [33]. Meanwhile, the fat content has the potential to delay gastric emptying, thereby slowing digestion and the absorption of glucose. Fat can also affect the interaction of plasma glucose, insulin, and GIP [33]. Therefore, the F2 with the highest protein content (13.33%) has a decreased glycemic response in the IAUC and has a low GI value (53.11).

Moreover, the high fiber content of F2 plays a role in slowing the rate of food digestion and inhibiting enzyme activity so that the digestive process, especially of starch, becomes slower and in turn lowers the blood glucose response [40]. Dietary fiber has a role in binding water and forming a matrix with carbohydrates in food to slow down the digestion and absorption of carbohydrates and keep blood glucose levels more stable [42]. Based on the research results, insoluble fiber has a stronger correlation than soluble fiber. Wolever [43] stated that the insoluble fiber contains uronic acid, which is found mainly in hemicellulose. The hemicellulose content in kidney beans was higher (20.2%) compared to other beans. It could inhibit starch digestion and reduce the glycemic response [44].

In this study, the resistant starch and phenolic content, both tannin and non-tannin phenol, did not correlate with the GI (Table 5). The low levels of resistant starch and phenol in the developed formulas seem to be the reason for this insignificant correlation (i.e., 2.8% and 7.5 mg of GAE/g, respectively) at F0 formulation (the highest), with no significant change after the kidney bean substitution for parameter-resistant starch and total phenol (Table 3). A meta-analysis by Afandi et al. [18] mentioned that the contents of phenol and resistant starch in beans are low-GI foods (i.e., 48.71 mg GAE/g and 21.27%, respectively).

### 3.4. GI of the Extruded Purple Sweet Potato as a Menu

Table 6 shows the values of the glycemic index and load when extruded food is consumed with milk. Based on the calculation results, the glycemic index is lower than just consuming the extruded tuber alone. The glycemic load of F0 consumed with milk is still in the high category (70.30), but it is lower than F0 alone (77.41). The GI of F1 decreased from the high category (74.68) to the moderate category (67.36) when consumed with milk. While F2, if consumed with milk, has a GI of 50.02, which is included in the low category, this is lower than when consumed without milk (53.11). Those decreases could be because milk is a source of animal protein that can affect the blood glucose response. Those results are in line with the research by Henry et al. [45], which states that adding milk to breakfast cereals can reduce the value of the meal’s GI. Milk protein has a strong insulinotropic effect. Casein and whey proteins are rich sources of leucine and phenylalanine, which have been shown to increase insulin secretion by stimulating cell function, thereby stimulating insulin production, which in turn lowers blood glucose response and results in lower GI values [45]. The developed, exposed tuber with 40% legume substitution provides a low GI (53) and even lower glycemic load (50) if consumed with milk. This profile provides a potential reason for the food to be consumed by pre-diabetic people. An index lower than 55 indicates that this extruded tuber might slowly release glucose into the blood, which mildly influences the pancreatic hormone.

### 3.5. Antioxidant Capacity of the Extruded Purple Sweet Potato

Beyond those excellent nutrition profiles after the kidney bean substitution in the developed EPSP, especially on the protein and fiber content and the GI value; however, the analysis on the antioxidant capacity of those products still needs to provide a satisfactory result. The extrusion process significantly reduces anthocyanin’s content by about 48% (Figure 4), i.e., from 1073 to 560 mg/kg for PSPF and F2, respectively. The same figure for the antioxidant capacity is indicated by IC_50_ and AEAC (Ascorbic Acid Equivalent Antioxidant Capacity) (Figure 4). The extrusion process lowers the antioxidant ability of the EPSP by up to 19%, i.e., from 674.08 to 548.34 mg vitamin C/100 g. The amount of antioxidant-containing substance needed to scavenge 50% of the initial radicals increased to 48%, i.e., from 944.99 to 1325.96 ppm. Indeed, the raw PSP has a strong antioxidant capacity, i.e., IC_50_ of 65.35 ppm [8], or equal to 1581.8 vitamin C/100 g [46].

The high thermal processes during the drying and extrusion, at 120 and 50 °C, respectively, seem to be the main plausible reason for the reduction. Research by Suzery et al. (2020) [47] concluded that heating could reduce total anthocyanin levels and the antioxidant capacity. Heating at 80 °C for 60 min reduced the total anthocyanin content by 55.79%, and the IC_50_ value increased from 42.81 to 129.61 ppm [47]. Other research reported that extrusion significantly reduced antioxidant activity and total phenol in all extruded samples by 60–68% compared to unprocessed foods [48,49]. High temperatures during extrusion can change the molecular structure of phenolic compounds and reduce their chemical reactivity due to a certain degree of polymerization, which causes a lack of antioxidant properties [48]. Some novel processing techniques like steam explosion [50] and low thermal and vacuum drying methods during the drying process could be an alternative solution to preserve the valuable native antioxidant capacity of purple sweet potato.

## 4. Conclusions

In conclusion, the substitution of 40% of legumes in the extruded purple sweet potato has proven to be a promising approach for enhancing protein and fiber contents of the final product by up to 13.33% and 16.31%, respectively. Furthermore, this supplementation has resulted in a desirable glycemic index profile, making the extruded tuber a potentially valuable food source with high fiber and protein contents that may be suitable for consumption by both normal and pre-diabetic individuals. However, to fully unlock the potential health benefits of purple sweet potatoes, innovative flour and extrusion processing methods are necessary to preserve the native antioxidant capacity of the plant. By doing so, it may be possible to create a new generation of extruded food products that offer enhanced nutritional value and promote better health outcomes for consumers.

## Figures and Tables

**Figure 1 foods-12-01525-f001:**
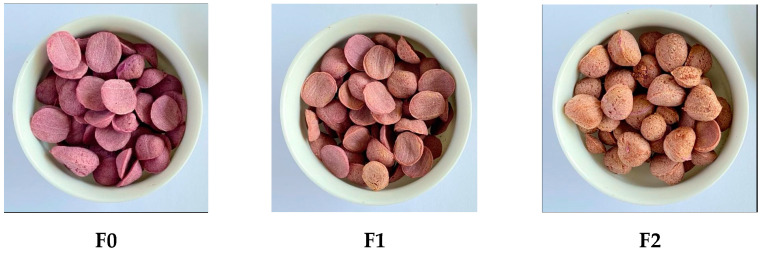
The visual appearance of extruded purple sweet potato with various proportions of kidney bean flour according to the formulation given in Table 1.

**Figure 2 foods-12-01525-f002:**
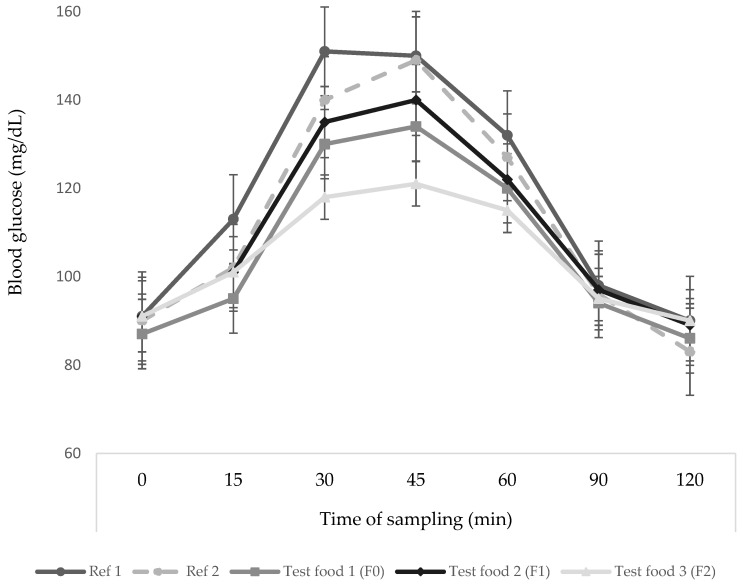
Glycemic response (means ± SE) of subjects after consuming glucose as reference food (Ref 1 and Ref 2) and F0, F1, and F2 as test foods. Ref 1, First replication of the reference food; Ref 2, Second replication of the reference food; F, Formula according to the formulation given in Table 1; F0, Formula with 0% substitution of kidney bean flour; F1, Formula with 30% substitution of kidney bean flour; F2, Formula with 40% substitution of kidney bean flour.

**Figure 3 foods-12-01525-f003:**
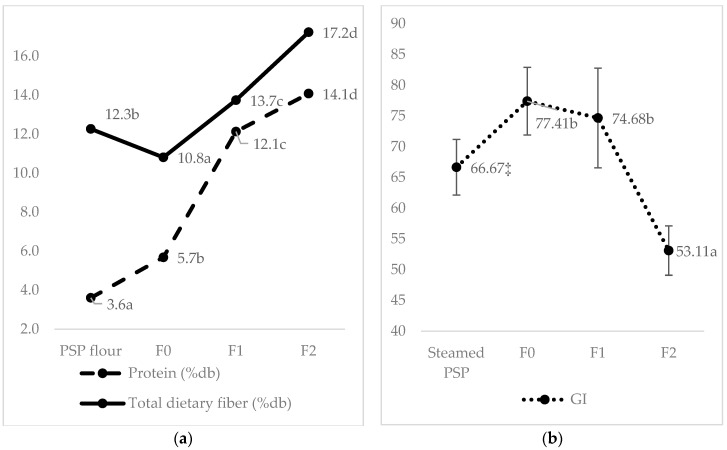
Content of protein and total dietary fiber (%db) (means ± SD) (**a**) and GI (means ± SEM) (**b**) are affected by processing type, i.e., drying for PSP flour, wet heating for steamed PSP, and extruded PSP with different levels of kidney bean substitution (0% for F0, 30% for F1, and 40% for F2). ^a–d^, Different superscript letters in each column are significantly different at *p* < 0.05, analyzed using one-way ANOVA and the Duncan post hoc test; ‡ Data was adopted from previous unpublished research that used the same methodology and the same PSP variant (ayamurazaki); PSP, Purple Sweet Potato.

**Figure 4 foods-12-01525-f004:**
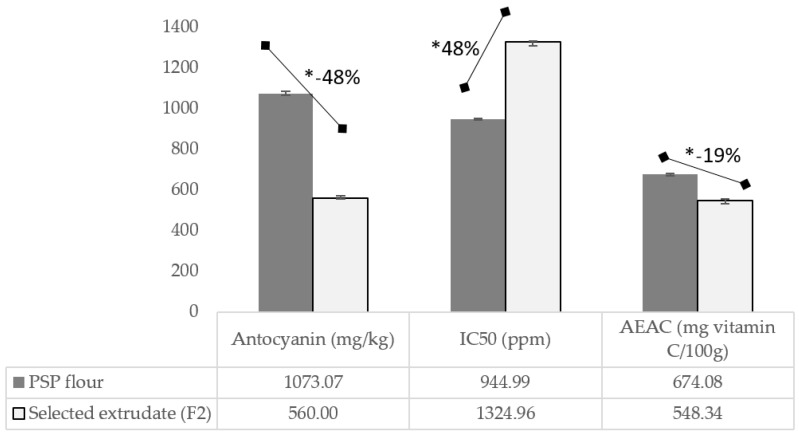
Profile of anthocyanin content, IC_50_, and AEAC of PSP flour and extruded PSP flour (selected formula: F2). * Significantly different at *p* < 0.05, analyzed using a *t*-test; F2, Formula with 40% substitution of kidney bean flour; IC_50_, the concentration of an antioxidant-containing substance required to scavenge 50% of the initial DPPH radicals; PSP, purple sweet potato; AEAC, Ascorbic Acid Equivalent Antioxidant Capacity.

**Table 1 foods-12-01525-t001:** Formulation of extruded purple sweet potatoes with three levels of kidney bean flour substitution (0, 30, and 40%).

Ingredients (g, %)	F0 (100% PSPF)	F1 (70% PSPF, 30% KBF)	F2 (60% PSPF, 40% KBF)
Purple sweet potato flour (PSPF)	2100 (76)	1470 (53)	1260 (46)
Kidney bean flour (KBF)	0 (0)	630 (23)	840 (30)
Rice flour	340 (12.46)	340 (12.46)	340 (12.46)
Cornstarch	45 (1.6)	45 (1.6)	45 (1.6)
Powdered milk	25 (0.9)	25 (0.9)	25 (0.9)
Palm oil	50 (1.8)	50 (1.8)	50 (1.8)
Water	200 (7.2)	200 (7.2)	200 (7.2)
Emulsifier	1 (0.046)	1 (0.046)	1 (0.046)

PSPF, Purple Sweet Potato Flour; KBF, Kidney Bean Flour; F0, Formula made from 100% PSPF; F1, Formula made from 70% PSPF and 30% KBF; F2, Formula made from 60% PSPF and 40% KBF.

**Table 2 foods-12-01525-t002:** Subject characteristics for glycemic index assessment.

Subject Code	Gender	Age (Years)	Weight (kg)	Height (cm)	BMI ^2^ (kg/m^2^)
001	Man	21	54.5	155.5	22.54
002	Man	21	55.0	160.5	21.35
003	Man	21	50.1	163.9	18.65
004	Man	21	60.8	169.4	21.19
005	Man	23	50.8	165.1	18.64
006	Man	23	54.8	160.9	21.17
007	Man	23	51.5	155.8	21.22
008	Woman	23	46.1	147.1	21.30
009	Woman	22	53.2	160.9	20.55
010	Woman	21	54.6	160.7	21.14
011	Woman	21	54.1	155.0	22.52
012	Woman	22	51.3	157.5	20.68
013	Woman	22	57.1	155.2	23.71
Mean ± SD ^1^		21.85 ± 0.90	53.38 ± 3.60	159.04 ± 5.62	21.13 ± 1.40

^1^ SD, Standard Deviation; ^2^ BMI, Body Mass Index.

**Table 3 foods-12-01525-t003:** Chemical compositions of PSPF, KBF, and the developed formulas (F0, F1, and F2).

Chemical Composition	Unit	PSPF	KBF	F0	F1	F2
Protein	%wb	3.50 ± 0.05 ^a^	16.40 ± 0.03 ^e^	5.36 ± 0.01 ^b^	11.48 ± 0.05 ^c^	13.33 ± 0.05 ^d^
Fat	%wb	0.40 ± 0.01 ^a^	1.60 ± 0.01 ^e^	0.50 ± 0.01 ^b^	0.76 ± 0.04 ^c^	0.87 ± 0.03 ^d^
Dietary fiber	%wb	11.90 ± 0.01 ^b^	13.12 ± 0.05 ^c^	10.20 ± 0.09 ^a^	13.01 ± 0.04 ^c^	16.31 ± 0.03 ^d^
Water soluble fiber	%wb	7.74 ± 0.01 ^c^	5.59 ± 0.01 ^b^	6.17 ± 0.53 ^b^	4.10 ± 0.02 ^a^	3.24 ± 0.08 ^a^
Water insoluble fiber	%wb	5.69 ± 0.01 ^a^	8.79 ± 0.01 ^b^	10.79 ± 0.09 ^c^	10.33 ± 0.07 ^c^	13.75 ± 0.05 ^d^
Resistant starch	%wb	2.32 ± 0.25 ^b^	1.40 ± 0.03 ^a^	2.80 ± 0.01 ^b^	2.47 ± 0.04 ^b^	2.26 ± 0.04 ^b^
Total phenol	mg GAE/g	10.30 ± 0.01 ^d^	1.20 ± 0.01 ^c^	7.50 ± 0.04 ^b^	7.20 ± 0.08 ^b^	6.80 ± 0.09 ^b^
Tannin phenol	mg GAE/g	3.62 ± 0.12 ^b^	0.10 ± 0.14 ^a^	4.84 ± 0.46 ^c^	2.55 ± 0.03 ^a^	2.12 ± 0.04 ^a^
Non-tannin phenol	mg GAE/g	6.60 ± 0.06 ^e^	1.09 ± 1.92 ^a^	2.66 ± 0.03 ^b^	4.62 ± 0.03 ^c^	4.74 ± 0.04 ^d^

PSPF, Purple Sweet Potato Flour; KBF, Kidney Bean Flour; F0, Formula made from 100% PSPF; F1, Formula made from 70% PSPF and 30% KBF; F2, Formula made from 60% PSPF and 40% KBF; different superscript letters in each row are significantly different at *p* < 0.05, analyzed using one-way ANOVA and the Duncan post hoc test; wb, wet basis; GAE, Gallic Acid Equivalents.

**Table 4 foods-12-01525-t004:** Glycemic index (GI ± SEM) of extruded purple sweet potato and kidney bean.

Food Samples	Time of Sampling (min)							IAUC	GI ± SEM
	0	15	30	45	60	90	120		
Ref 1	91 ± 2.4 ^a^	113 ± 3.7 ^b^	151 ± 5.5 ^c^	150 ± 5.6 ^b^	132 ± 6.4 ^b^	98 ± 6.4 ^a^	90 ± 4.7 ^a^	3420.11 ± 540.6 ^b^	
Ref 2	90 ± 2.4 ^a^	102 ± 2.4 ^a^	140 ± 4.0 ^bc^	149 ± 6.0 ^b^	127 ± 4.8 ^ab^	96 ± 4.3 ^a^	83 ± 3.7 ^a^	2918.33 ± 240.6 ^b^	
F0	87 ± 1.3 ^a^	95 ± 3.0 ^a^	130 ± 5.0 ^ab^	134 ± 5.3 ^ab^	120 ± 4.5 ^ab^	94 ± 3.4 ^a^	86 ± 3.4 ^a^	2456.22 ± 288.6 ^ab^	77.41 ± 5.5 ^b^
F1	91 ± 0.8 ^a^	101 ± 5.5 ^a^	135 ± 7.2 ^b^	140 ± 7.4 ^b^	122 ± 6.4 ^ab^	97 ± 5.1 ^a^	89 ± 2.0 ^a^	2536.97 ± 446.6 ^ab^	74.68 ± 8.1 ^b^
F2	91 ± 2.5 ^a^	101 ± 3.6 ^a^	118 ± 4.6 ^a^	121 ± 3.6 ^a^	115 ± 4.4 ^a^	95 ± 4.1 ^a^	90 ± 3.1 ^a^	1737.70 ± 211.5 ^a^	53.11 ± 4.0 ^a^

^a–c^, Different superscript letters in each column are significantly different at *p* < 0.05, analyzed using one-way ANOVA and the Duncan post hoc test; IAUC, Incremental Area Under the Curve; GI, Glycemic Index; SEM, Standard Error Means; Ref 1, First replication of the reference food; Ref 2, Second replication of the reference food; F, Formula according to the formulation given in Table 1; F0, Formula with 0% substitution of kidney bean; F1, Formula with 30% substitution of kidney bean flour; F2, Formula with 40% substitution of kidney bean flour.

**Table 5 foods-12-01525-t005:** Correlation between chemical compositions and the glycemic index.

Chemical Composition	Glycemic Index	
	r	*p*-Value
Protein	−0.827	0.042 *
Fat	−0.855	0.030 *
Dietary fiber	−0.962	0.002 *
Water soluble fiber	0.897	0.015 *
Water insoluble fiber	−0.930	0.007 *
Resistant starch	0.857	0.129
Total phenol	0.686	0.133
Tannin phenol	0.762	0.078
Non-tannin phenol	−0.716	0.110

r, Pearson correlation; *, Significantly related at *p* < 0.05.

**Table 6 foods-12-01525-t006:** Glycemic index and glycemic load of extruded purple sweet potatoes.

Meal	Serve (g)	Available Carbohydrate (g)	Carbohydrate Contribution (%)	Glycemic Index (GI)	Meal Glycemic Index	Meal Glycemic Load
F0	35	26.40	81.48	77.41	63.07	20.44
Pasteurized milk *	125	6.00	18.52	39.00	7.22	2.34
Total Meal GI of F0 with milk	160	32.40	100.00	116.41	70.30	22.78
F1	35	23.23	79.47	74.68	59.35	17.35
Pasteurized milk *	125	6.00	20.53	39.00	8.01	2.34
Total Meal GI of F1 with milk	160	29.23	100.00	113.68	67.36	19.69
F2	35	21.37	78.08	53.11	41.47	11.35
Pasteurized milk *	125	6.00	21.92	39.00	8.55	2.34
Total Meal GI of F0 with milk	160	27.37	100.00	92.11	50.02	13.69

* Adopted from Philippou (2017) [24]; F0, Formula made from 100% purple sweet potato flour; F1, Formula with 30% substitution of kidney bean flour; F2, Formula with 40% substitution of kidney bean flour.

## Data Availability

Data is contained within the article.

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
