# Peer review of "Kidney Bean Substitution Ameliorates the Nutritional Quality of Extruded Purple Sweet Potatoes: Evaluation of Chemical Composition, Glycemic Index, and Antioxidant Capacity"

_foods, 2023, doi:10.3390/foods12071525_

Round 1
Reviewer 1 Report
The manuscript that I am reviewing is interesting and brings new information about extruded food technology.
The introduction is prepared correctly and introduces the issues of obesity and diabetes well, but I lacked technological information on the production of extruded food. What is the range currently on the market? Is the proposed product dedicated to any of the population groups? Is the proposed snack supposed to be so-called functional food? I think it's worth adding these informations.
The 'material and methods' part is described adequately, but I found a few mistakes. Please correct them:
- in Table 1, in brackets next to 'Kidney bean flour', the designation KBF should be placed, not PSPF;
- under Table 1 (lines 80-82) the authors explain what the abbreviation 'EPSP' stands for, but this abbreviation is not used in this table'
-I have the impression that the same fragment was copied to the next lines (85-87) as the caption of figure 1. It is redundant, because these markings are not used in this figure;
Line 90: I do not understand what '35 mesh' means. Please explain it in the text, or remove the text from brackets.
Line 106: same as above.
Line 137: I wonder why 25g of glucose was chosen to determine the glycemic index and not 50g? The authors explain that the tested foods have a very light specific gravity, but by definition the GI measurement is based on the glycemic response in the blood after a 12-hour fast and eating a food with approximately 50 g of available carbohydrates.
Line 162: What is 'milk21'?
Results and Discussion
Lines 183-185 are redundant.
Lines 198-200: The abbreviation 'EPSP' is not used in the first row of Table 3, so there is no need to explain it below the table.
Lines 227-230: This is not a proper caption under the figure 2. Explain what 'Ref 1' and 'Ref 2' mean.
Lines 253-255, 263-265, 324-326, 343-345, : Please correct the explanation below the tables and figures.
Table 5: Average values given? Why are there no statistics?
Conclusion: Conclusions could to be improved.
References: I am not sure if the correct citation format required by Foods was used.
Author Response
Response letter
This letter is referring to the revision of the manuscript entitled:
“Kidney Bean Substitution Ameliorates the Nutritional Quality of Extruded Purple Sweet Potato: Evaluation of Chemical Composition, Glycemic Index, and Antioxidant Capacity” with Manuscript ID: foods-2305887, written by Eny Palupi, Nira Delina, Naufal M. Nurdin, Hana F. Navratilova, Rimbawan Rimbawan and Ahmad Sulaeman.
General comment
Dear Dr. Palupi,
Thank you again for your manuscript submission:
Manuscript ID: foods-2305887
Type of manuscript: Article
Title: Kidney Bean Substitution Ameliorates the Nutritional Quality of Extruded Purple Sweet Potato: Evaluation of Chemical Composition, Glycemic Index, and Antioxidant Capacity
Authors: Eny Palupi *, Nira Delina, Naufal M Nurdin, Hana F Navratilova, Rimbawan Rimbawan, Ahmad Sulaeman
Received: 10 March 2023
E-mails: enypalupi@apps.ipb.ac.id, nira_1620@apps.ipb.ac.id,
naufal@apps.ipb.ac.id, hana.fitria@apps.ipb.ac.id, rimbawan@apps.ipb.ac.id, asulaema06@gmail.com
Submitted to section: Food Engineering and Technology,
https://www.mdpi.com/journal/foods/sections/Food_Processing_Technology
Current Practice and Future Directions of Application of Puffed/Extruded Technologies in Food
https://www.mdpi.com/journal/foods/special_issues/Puffed_Extruded_food
Your manuscript has been reviewed by experts in the field. Please find your manuscript with the referee reports at this link:
https://susy.mdpi.com/user/manuscripts/resubmit/6bdbeb198e902773dbf28d29ceb683de
--------------------------------
We may receive other review reports for your manuscript later and we will send the comments to you immediately. However, if we have not received the report before the revision deadline, we would cancel the review request.
--------------------------------
(I) Please revise your manuscript according to the referees’ comments and upload the revised file within 5 days.
(II) Please use the version of your manuscript found at the above link for your revisions.
(III) Please check that all references are relevant to the contents of the manuscript.
(IV) Any revisions made to the manuscript should be marked up using the “Track Changes” function if you are using MS Word/LaTeX, such that changes can be easily viewed by the editors and reviewers.
(V) Please provide a short cover letter detailing your changes for the editors’ and referees’ approval.
If one of the referees has suggested that your manuscript should undergo extensive English revisions, please address this issue during revision. We propose that you use one of the editing services listed at
https://www.mdpi.com/authors/english or have your manuscript checked by a native English-speaking colleague.
Please do not hesitate to contact us if you have any questions regarding the revision of your manuscript or if you need more time. We look forward to hearing from you soon.
Kind regards,
Ms. Lisa Sun
Section Managing Editor, MDPI Beijing
E-Mail: lisa.sun@mdpi.com
General response
Dear the Editors and Referees of Foods,
thank you very much for the favorable feedback about our manuscript and for giving us a chance to carry out revision along the comments given by the reviewers. Below please find our point-by-point responses to the comments from the reviewers. All line numbers given refer to the revised version of the manuscript. All the changes are marked with track changes for revision in the manuscript.
Date: 25.03.2023
With kind regards, on behalf of all authors,
Eny Palupi
Specific comments for Referee 1
Comment #1.
The manuscript that I am reviewing is interesting and brings new information about extruded food technology.
The introduction is prepared correctly and introduces the issues of obesity and diabetes well, but I lacked technological information on the production of extruded food. What is the range currently on the market? Is the proposed product dedicated to any of the population groups? Is the proposed snack supposed to be so-called functional food? I think it's worth adding these informations.
Response: Thank you very much for the favorable input and significant recommendation. The important information related to the current issue of extruded food has been added to the text. The target population and the purpose of the product development also have been described in the introduction part as follow:
Extruded foods have been a topic of discussion in the food industry for many years and continue to be a popular and convenient food choice for many consumers. Some concerns have been raised regarding the nutritional quality and safety of extruded foods [8,9]. Extruded foods are made by processing a mixture of ingredients, such as grains and legumes, under high pressure and temperature to create a dough-like material that is then pushed through a die to give it a specific shape. One concern with extruded foods is that the high heat and pressure used during processing can cause a loss of nutrients, especially vitamins and bioactive compounds [8,9]. Additionally, some studies have suggested that certain compounds formed during the extrusion process may have negative health effects. Some alternative approach to handle this issue is using low temperature extrusion and legumes substitution on the mixture [10,11]. (Line 45–55)
The final purpose of this research is to develop a nutritious product that is well-suited for promoting and maintaining the nutritional status and health of the community. (Line 71–73)
Comment #2
The 'material and methods' part is described adequately, but I found a few mistakes. Please correct them:
- in Table 1, in brackets next to 'Kidney bean flour', the designation KBF should be placed, not PSPF;
- under Table 1 (lines 80-82) the authors explain what the abbreviation 'EPSP' stands for, but this abbreviation is not used in this table'
Response: Many thanks for pointing out the mistakes. It has been revised accordingly (Table 1)
Comment #3
-I have the impression that the same fragment was copied to the next lines (85-87) as the caption of figure 1. It is redundant, because these markings are not used in this figure;
Response: Thank you very much for the remarks on the mistakes. The necessary revisions have been made accordingly (Figure 1).
Comment #4
Line 90: I do not understand what '35 mesh' means. Please explain it in the text, or remove the text from brackets.
Line 106: same as above.
Response: Thank you for the suggestion. It has been removed as suggested (Line 109, 125).
Comment #5
Line 137: I wonder why 25g of glucose was chosen to determine the glycemic index and not 50g? The authors explain that the tested foods have a very light specific gravity, but by definition the GI measurement is based on the glycemic response in the blood after a 12-hour fast and eating a food with approximately 50 g of available carbohydrates.
Response: Thank you very much for the feedback. Based on the ISO 26642-2010 about the Determination of the glycaemic index (GI), in the Chapter 2.2 about carbohydrate portion, it is stated that the GI values could be determined by the weighed portion of food containing either 50 g of glycaemic carbohydrate or, if the portion size is unreasonably large, 25 g of glycaemic carbohydrate may be used. The evaluated samples have a very light specific gravity, so it has high density (volume per weight). This high volume per weight made the 50g glycaemic carbohydrate equal to 82g product (about two bowls) which is unreasonable to be consumed within 12 minutes. Therefore the study used 25g of available carbohydrate, instead of 50g of available carbohydrate. This detail explanation has been added in the material and methods part (Lines 175–193).
Comment #6
Line 162: What is 'milk21'?
Response: Thank you for the remark on the mistyped. It has been revised accordingly (Line 212).
Comment #7
Results and Discussion
Lines 183-185 are redundant.
Response: Many thanks you for the suggestion. It has been removed as suggested (Line 263).
Comment #8.
Lines 198-200: The abbreviation 'EPSP' is not used in the first row of Table 3, so there is no need to explain it below the table.
Response: Thank you very much for the suggestion. It has been removed as suggested (Table 3).
Comment #9.
Lines 227-230: This is not a proper caption under the figure 2. Explain what 'Ref 1' and 'Ref 2' mean.
Response: Thank you for the remark. It has been revised accordingly (Figure 2).
Comment #10.
Lines 253-255, 263-265, 324-326, 343-345: Please correct the explanation below the tables and figures.
Response: Thank you very much for the suggestion. It has been corrected accordingly (Table 4, 6 and Figure 3, 4).
Comment #11.
Table 5: Average values given? Why are there no statistics?
Response: Thank you very much for the remark. The value of meal Glycemic Index (Table 6) were calculated mathematically using the weighted average GI values for each individual ingredient based on its contribution to the overall available carbohydrate content. This detail explanation has been added in the material and methods Line 212–214 This method was adopted from Philippou (2017) [24]. Since the values were predicted mathematically, only a single value was produced for each menu/meal and could not be analyzed statistically.
Comment #12.
Conclusion: Conclusions could to be improved.
Response: Thank you very much for the suggestion. The conclusions have been improved as follow:
“In conclusion, the substitution of 40% of legumes in the extruded purple sweet potato has proven to be a promising approach to enhance the protein and fiber content of the final product by up to 13.33% and 16.31%, respectively. Furthermore, this supplementation has resulted in a desirable glycemic index profile, making the extruded tuber a potentially valuable source of high-fiber and high-protein staple food that may be suitable for consumption by both normal and pre-diabetic individuals. However, to fully unlock the potential health benefits of the purple sweet potato, an innovative flour and extrusion processing method is necessary to preserve the native antioxidant capacity of the plant. By doing so, it may be possible to create a new generation of extruded food products that offer enhanced nutritional value and promote better health outcomes for consumers.” (Line 534–568).
Comment #13.
References: I am not sure if the correct citation format required by Foods was used.
Response: Thank you very much for the remark. The format has been revised based on the Foods guideline provided. (Line 579–694).
Specific comments for Referee 2
Comment #1.
The experiments had some nice results; however, I have some questions and suggestions:
- Lines 61 - 63: These should be removed and placed in the Acknowledgment section if deemed necessary.
Response: Thank you very much for the suggestion. Since the method applied in this research is involved the human specimen, the research needs ethical permission, which is stated in the beginning of the material and methods part.
Comment #2.
- Lines 183 - 185: should be removed.
Response: Thank you very much for the favorable remark. The unnecessary sentences have been removed (Lines 263).
Comment #3.
- Lines 177 - 181: Please provide detailed information on the statistics software used.
Response: Thank you very much for the favorable remark. The detailed information on the statistics software used have been added (Lines 261–262).
Comment #4.
- For GI assessment, with a limited number of participants, it can be challenging to differentiate true effects from random variation or noise. As a result, the findings may not accurately represent the broader population, and it may not be feasible to draw reliable conclusions or make informed decisions based on them.
Response: Thank you very much for the remark. The determination of GI values in our research referred to the international standard for Determination of the glycaemic index (GI) ISO 26642-2010 page 3, which mention that the GI should be determined based on the average glucose response of a minimum of 10 healthy subjects. This has been mentioned in material and methods part Lines 153–155 that the GI value in this study was assessed in vivo involved 13 participating healthy subjects.
Comment #5.
- Lines 165 - 167: the sentence should be improved.
Response: Thank you very much for the favorable remark. The sentences have been rewritten to make it understandable (Lines 165 – 167).
The glycemic index (GI) is a system that ranks foods based on how they affect blood sugar levels. The GI values are classified into three categories: high GI (≥70), medium GI (55–70), and low GI (≤55). Another metric used to measure the impact of a food on blood sugar levels is the glycemic load (GL), which takes into account both the GI of the food and the amount of carbohydrates in a typical serving. The GL categories are defined as follows: high GL (>20), medium GL (11–19), and low GL (<10) (ISO, 2010). The formula for GL is [GI of food x Carbohydrate content (g) in one serving size] x 100. (Lines 214–221)
Specific comments for Referee 3
Comment #1.
The manuscript deals with an interesting and important topic, but it is of a low scientific level. Simple, basic analytical methods were used. a lot of attention was focused on plofienols and their role improving health functions in the body, but in this case their profile was not determined. It would have been valuable to compare the profile before and after the process.
Response: Thank you very much for pointing out this issue. The content of tannin, non-tannin, and total phenol was presented in the Table 3. In this study, the resistant starch and phenolic content, both tannin and non-tannin phenol, do not correlate with the GI (Table 5). This has been proved statistically and presented in the Table 4. The low level of resistant starch and phenol of the developed formulas seems to be the reason for this insignificant correlation, i.e. 2.8% and 7.5 mg GAE/g, respectively, at F0 formulation (the highest) with no significant change after the kidney bean substitution (Table 3). A meta-analysis by Afandi et al. [14] mentioned that the contents of phenolic and resistant starch of beans with a low GI are 48.71 mg GAE/g and 21.27 %, respectively (Lines 450–465).
Comment #2.
Line - residue template 182-185 delete
Response: Thank you, it has been removed as suggested (Line 1).
Comment #3.
Line - 209-215: Please explain the reason for the increase in non-tannin phenol in extrudates with KBF, compared to PSPF. Please complete whether the determined content of non-tannin phenol in extrudates with KBF in the amount taken with this type of product is relevant for the described health effects. Discussion
Please rethink your discussion if you are sure it relates to the obtained results.
Response: Thank you for addressing this critical issue. The findings indicated a significant increase in the content of non-tannin phenols in extruded PSP with increased KBF substitution. This phenomenon is hypothesized to be due to the conversion of tannin phenols into non-tannin phenols under the high temperature and intense friction conditions during the extrusion process [50]. The process appeared to alter the characteristic of the key functional groups of the phenolic compound into non-tannin phenols. This process seems to reduce their ability to bind to proteins, as evidenced by the PPVP test. The conversion of tannin phenols into non-tannin phenols may be attributed to the cleavage of the hydrolysable tannins, which are more susceptible to thermal degradation [50]. Therefore, the extrusion process can lead to changes in the composition and functional properties of PSP, as well as alterations in its bioactive components. The increase of KBF substitution seems enhanced the above reaction. However, the amount of tannins in the extruded tuber in this study has not been able to have an effect on the GI value. A complete discussion regarding this correlation is thoroughly reviewed in subsection 3.3. (Line 292–326)
Comment #4.
Conclusion
please support with the result.
Response: Thank you very much for the suggestion. the conclusion part has been rewrote and improved as follow:
“In conclusion, the substitution of 40% of legumes in the extruded purple sweet potato has proven to be a promising approach to enhance the protein and fiber content of the final product by up to 13.33% and 16.31%, respectively. Furthermore, this supplementation has resulted in a desirable glycemic index profile, making the extruded tuber a potentially valuable source of high-fiber and high-protein staple food that may be suitable for consumption by both normal and pre-diabetic individuals. However, to fully unlock the potential health benefits of the purple sweet potato, an innovative flour and extrusion processing method is necessary to preserve the native antioxidant capacity of the plant. By doing so, it may be possible to create a new generation of extruded food products that offer enhanced nutritional value and promote better health outcomes for consumers.” (Line 534–568)

Reviewer 2 Report
The experiments had some nice results; however, I have some questions and suggestions:
- Lines 61 - 63: These should be removed and placed in the Acknowledgment section if deemed necessary.
- Lines 183 - 185: should be removed.
- Lines 177 - 181: Please provide detailed information on the statistics software used.
- For GI assessment, with a limited number of participants, it can be challenging to differentiate true effects from random variation or noise. As a result, the findings may not accurately represent the broader population, and it may not be feasible to draw reliable conclusions or make informed decisions based on them.
- Lines 165 - 167: the sentence should be improved.
Author Response

(The authors gave the same response as above.)

Reviewer 3 Report
The manuscript deals with an interesting and important topic, but it is of a low scientific level. Simple, basic analytical methods were used. a lot of attention was focused on plofienols and their role improving health functions in the body, but in this case their profile was not determined. It would have been valuable to compare the profile before and after the process.
Line - residue template 182-185 delete
Line - 209-215:
Please explain the reason for the increase in non-tannin phenol in extrudates with KBF, compared to PSPF. Please complete whether the determined content of non-tannin phenol in extrudates with KBF in the amount taken with this type of product is relevant for the described health effects.
Discussion
Please rethink your discussion if you are sure it relates to the obtained results.
Conclusion
please support with the result.
Author Response

(The authors gave the same response as above.)

Round 2
Reviewer 1 Report
Dear Authors,
Thank you for considering my suggestions and improving the manuscript as well as comprehensively answering my doubts.